# Edge-rich molybdenum disulfide tailors carbon-chain growth for selective hydrogenation of carbon monoxide to higher alcohols

Jingting Hu[1,2,7], Zeyu Wei[1,3,7], Yunlong Zhang[1,3], Rui Huang[1], Mingchao Zhang[2], Kang Cheng [2], Qinghong Zhang [2], Yutai Qi[1,2], Yanan Li[1,2], Jun Mao[1,2], Junfa Zhu [4], Lihui Wu[4], Wu Wen[4], Shengsheng Yu[4], Yang Pan [4], Jiuzhong Yang [4], Xiangjun Wei[5], Luozhen Jiang[6], Rui Si [6], Liang Yu [1,3] ✉, Ye Wang [2] ✉ & Dehui Deng [1,2,3] ✉

Selective hydrogenation of carbon monoxide (CO) to higher alcohols ($C_{2+}OH$) is a promising non-petroleum route for producing high-value chemicals, in which precise regulations of both C-O cleavage and C-C coupling are highly essential but remain great challenges. Herein, we report that highly selective CO hydrogenation to $C_{2-4}OH$ is achieved over a potassium-modified edge-rich molybdenum disulfide ($MoS_2$) catalyst, which delivers a high CO conversion of 17% with a superior $C_{2-4}OH$ selectivity of 45.2% in hydrogenated products at 240 °C and 50 bar, outperforming previously reported non-noble metal-based catalysts under similar conditions. By regulating the relative abundance of edge to basal plane, $C_{2-4}OH$ to methanol selectivity ratio can be overturned from 0.4 to 2.2. Mechanistic studies reveal that sulfur vacancies at $MoS_2$ edges boost carbon-chain growth by facilitating not only C-O cleavage but also C-C coupling, while potassium promotes the desorption of alcohols via electrostatic interaction with hydroxyls, thereby enabling preferential formation of $C_{2-4}OH$.

CO hydrogenation provides an attractive non-petroleum route for producing higher alcohols ($C_{2+}OH$), which are widely used as chemical feedstocks, fuels, fuel additives, and solvents, etc.[1–4]. In this reaction, formation of $C_{2+}OH$ typically involves two critical steps, i.e. (i) C-O cleavage of adsorbed CO (CO*) or alkoxyls ($CH_xO^*$, $x = 1–4$) to generate adsorbed C (C*) or alkyls ($CH_x^*$, $x = 1 – 3$) species and (ii) C-C coupling among the C*, $CH_x^*$, CO*, or $CH_xO^*$ for carbon-chain growth[5]. In these steps, C-O cleavage as the foregoing step for C-C coupling is of essential significance but also inevitably causes decomposition of the generated alcohols. Such a seesaw effect makes the selectivity of $C_{2+}OH$ hardly controllable.

[1]State Key Laboratory of Catalysis, Collaborative Innovation Center of Chemistry for Energy Materials, Dalian Institute of Chemical Physics, Chinese Academy of Sciences, Dalian 116023, China. [2]State Key Laboratory of Physical Chemistry of Solid Surfaces, Collaborative Innovation Center of Chemistry for Energy Materials, College of Chemistry and Chemical Engineering, Xiamen University, Xiamen 361005, China. [3]University of Chinese Academy of Sciences, Beijing 100049, China. [4]National Synchrotron Radiation Laboratory, University of Science and Technology of China, Hefei 230029, China. [5]Shanghai Synchrotron Radiation Facility, Shanghai Advanced Research Institute, Chinese Academy of Sciences, Shanghai 201204, China. [6]Shanghai Institute of Applied Physics, Chinese Academy of Sciences, Shanghai 201204, China. [7]These authors contributed equally: Jingting Hu, Zeyu Wei. ✉e-mail: lyu@dicp.ac.cn; wangye@xmu.edu.cn; dhdeng@dicp.ac.cn

A variety of catalysts have been explored for CO hydrogenation to $C_{2+}OH$, such as Rh-based catalysts[6,7], modified Cu-based methanol-synthesis catalysts[8–10], modified Fischer-Tropsch-synthesis (FT-synthesis) catalysts[11–14], and Mo-based catalysts[15–19]. Among these catalysts, the Rh-based catalysts are beneficial for ethanol synthesis but suffer from high price. The modified Cu-based methanol-synthesis catalysts usually give rise to a large proportion of methanol among alcohol products[8–10], since the C-C coupling for producing higher alcohols over the catalyst typically undergoes a sluggish aldol-type condensation mechanism. Modified FT-synthesis catalysts such as Co-based and Fe-based catalysts, exhibit high activity in CO hydrogenation because Co and Fe are classical active metals for C-O dissociation and C-C coupling[20]. But the irrepressible C-O cleavage and the random C-C coupling on the open surface of FT catalysts typically lead to wide distribution of carbon number in alcohols[12,13]. In addition, the above three catalysts usually suffer poor stability caused by phase separation, sintering, or sulfur poisoning under working conditions, which hinders their industrial application[6,21–25]. The Mo-based catalysts are well known in sulfur tolerance[21], but have relatively lower activity for C-O dissociation and C-C coupling, thus usually leading to suppressed carbon-chain growth with a high methanol selectivity and typically requiring high reaction temperatures (>300 °C) and harsh pressures (>80 bar) for the production of $C_{2+}OH$[15–19,26–28]. Though great progresses have been made for the development of catalysts for higher alcohols synthesis in previous works, the limited CO conversion (<10%) or $C_{2-4}OH$ selectivity (<40%) under mild reaction conditions is still insufficient and need to be improved. The development of highly efficient non-noble metal-based catalysts for the CO hydrogenation to $C_{2+}OH$ under mild conditions, requires precise regulations of active site in a nano-level to control both the C-O cleavage and C-C coupling, which are the keys for realizing controllable carbon-chain growth and selective formation of specific $C_{2+}OH$ but remain great challenges.

Herein, we report that on the basis of a nano channel-confined growth mechanism, uniform nano-arrays of potassium-modified edge-rich $MoS_2$ (ER-$MoS_2$-K) is prepared for highly selective CO hydrogenation to $C_{2-4}OH$. At relatively low temperature of 240 °C and low pressure of 50 bar, a high CO conversion of 17% is reached with a superior $C_{2-4}OH$ selectivity of 45.2% in hydrogenated products, surpassing that of previously reported non-noble metal-based catalysts under similar conditions. By reducing the lateral size of $MoS_2$ to enrich the edges for boosting the carbon-chain growth, the $C_{2-4}OH$ to methanol selectivity ratio is overturned from 0.4 to 2.2, and the selectivity of $C_{2-4}OH$ is almost 100% in the $C_{2+}OH$ products. In-situ characterizations combined with theoretical calculations reveal the sulfur vacancies (SVs) at the edge of the ER-$MoS_2$-K as the active sites for the CO hydrogenation to $C_{2-4}OH$, where not only the C-O cleavage of $CH_xO^*$ is facilitated for the generation of $CH_x^*$ intermediate, but also the subsequent C-C coupling between $CH_x^*$ and $CO^*$ is more favored for the growth of carbon-chain than deep hydrogenation to methane. The potassium promoter promotes the desorption of alcohols via electrostatic interaction with the hydroxyl of alcohols, thereby improving the selectivity toward $C_{2-4}OH$.

## Results
### Structure and performance of edge-rich $MoS_2$
Edge-rich $MoS_2$ (denoted as ER-$MoS_2$) was synthesized on the basis of nano channel-confined growth mechanism by using SBA-15 (a kind of ordered mesoporous silica) as a hard template to restrict the lateral

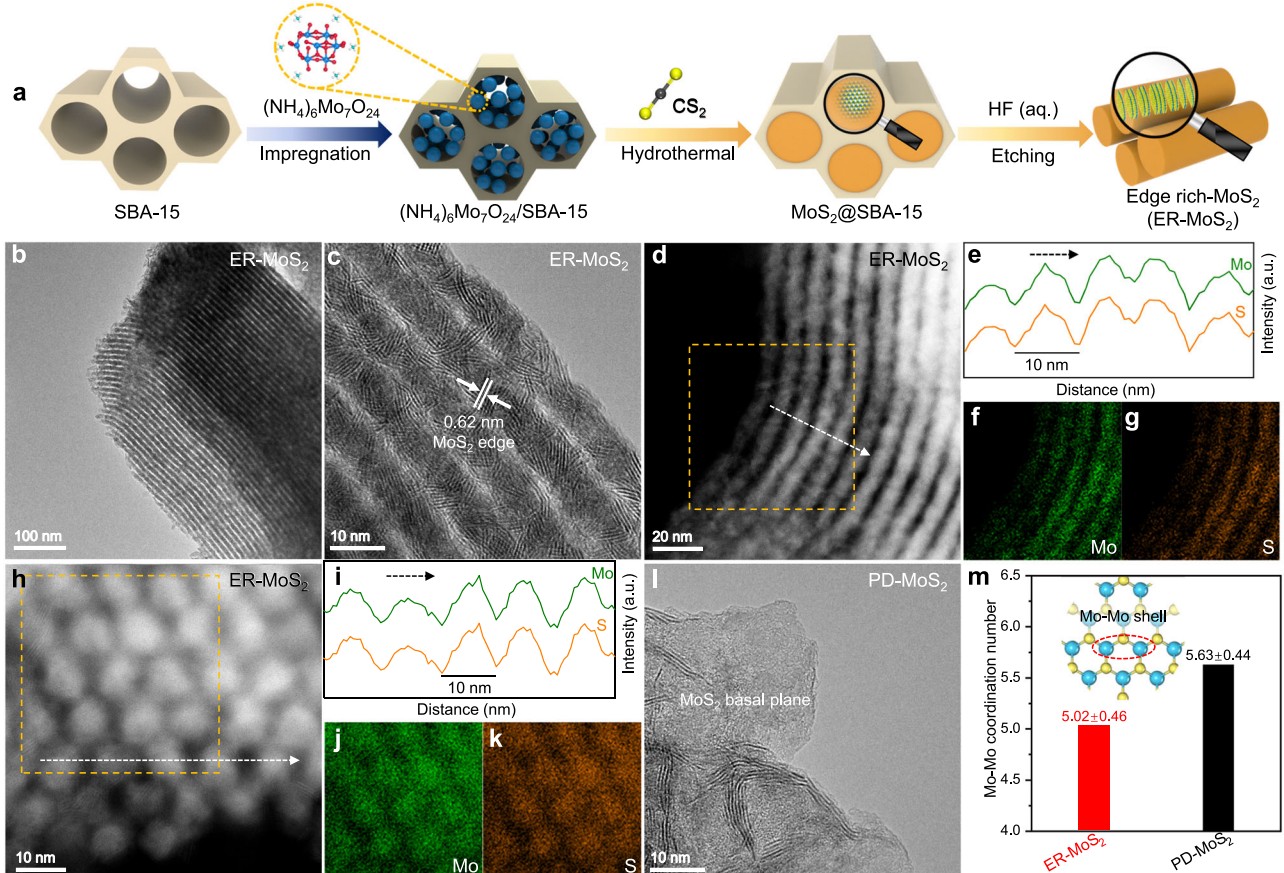

**Fig. 1 | Structural characterization of different $MoS_2$. a** Schematic illustration of the synthesis of ER-$MoS_2$. **b, c** Transmission electron microscopy (TEM) images of ER-$MoS_2$. **d–k** High-angle annular dark field scanning transmission electron microscopy (HAADF-STEM) images (**d, h** side view and top view, respectively), EDX line scans (**e, i**), and EDX mappings (**f, g, j, k**) of ER-$MoS_2$. **l,** TEM image of PD-$MoS_2$. **m** The coordination number of the Mo-Mo shell for ER-$MoS_2$ and PD-$MoS_2$.

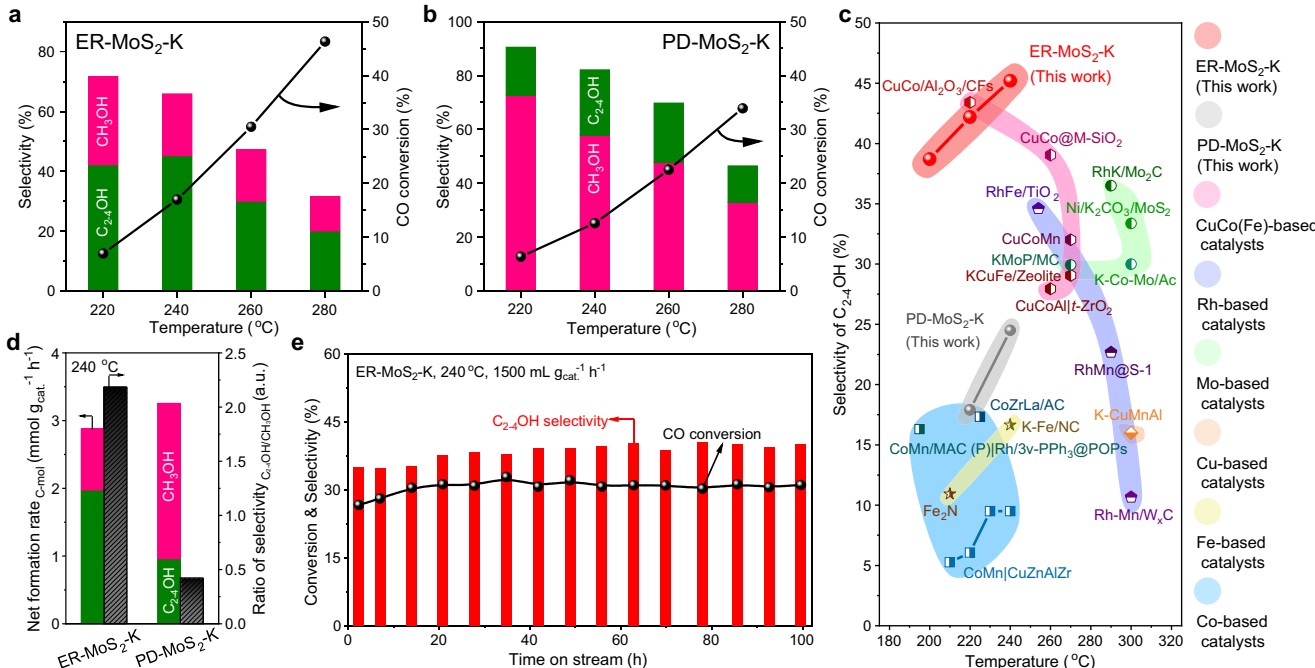

**Fig. 2 | Catalytic performances of different MoS₂-based catalysts. a, b** Catalytic performances of ER-MoS₂-K (**a**) and PD-MoS₂-K (**b**) at 3000 mL $g_{cat.}^{-1}$ $h^{-1}$. **c** Comparison in the selectivity of C₂₋₄OH over ER-MoS₂-K, PD-MoS₂-K and other catalysts reported in literatures (see Supplementary Table 3 for more details). **d** Net formation rate and distribution of alcohol products over ER-MoS₂-K and PD- MoS₂-K at 240 °C, 3000 mL $g_{cat.}^{-1}$ $h^{-1}$. **e** Stability test of the ER-MoS₂-K catalyst in CO hydrogenation reaction at 1500 mL $g_{cat.}^{-1}$ $h^{-1}$. The product selectivity was calculated on a CO₂-free basis. Catalysts were pretreated in-situ by H₂ at 300 °C for 1 h before reaction. Reaction activity tests were performed using a tubular fixed-bed reactor at 50 bar and a H₂/CO of 2.

growth of MoS₂ in the confined space of its mesoporous channels (Fig. 1a, Supplementary Fig. 1, and Supplementary Table 1). After removing the SBA-15 template, structural characterizations show that the obtained ER-MoS₂ consists of nanosheets with small lateral sizes and abundant edges assembled in nano-array morphology with uniform linear channels as derived from the template (Fig. 1b-k, Supplementary Fig. 2a). As shown in energy-dispersive X-ray (EDX) line scans of ER-MoS₂ (Fig. 1e, i), the diameter of an array is around 10 nm, corresponding to the pore diameter of SBA-15 (Supplementary Fig. 1). Basal plane-dominated MoS₂ (denoted as PD-MoS₂) with relatively large lateral sizes was also synthesized as a reference catalyst to investigate the roles of MoS₂ edge and basal plane in the CO hydrogenation reaction (Supplementary Fig. 2b). X-ray diffraction (XRD) patterns demonstrate that both ER-MoS₂ and PD-MoS₂ exhibit typical characteristics of hexagonal 2H-MoS₂ crystal (Supplementary Fig. 2c). The PD-MoS₂ with large lateral sizes mainly exposes the basal plane of MoS₂ sheets (Fig. 1l, Supplementary Fig. 2b), in sharp contrast with the ER-MoS₂ which mainly exposes edges. This is also reflected by the Fourier-transformed extended X-ray absorption fine structure (EXAFS) spectra, which show that the ER-MoS₂ has less Mo-Mo coordination than the PD-MoS₂ (Fig. 1m and Supplementary Fig. 2d, e), further confirming that the ER-MoS₂ possesses more edge sites than the PD-MoS₂[29,30].

Potassium was further introduced as a promoter for MoS₂ but without affecting the overall pore size distributions and structures of MoS₂ by using an impregnant method with potassium carbonate as the source of potassium (Supplementary Fig. 3, Supplementary Table 2). Catalytic performances of the potassium-modified MoS₂ (denoted as MoS₂-K) catalysts for CO hydrogenation were evaluated in a fixed-bed reactor (Fig. 2a, b). Over the ER-MoS₂-K catalyst with an optimized K/ Mo mole ratio of 0.2 (Supplementary Fig. 4), C₂₋₄OH are always the primary alcohol product at different temperatures from 220 to 280 °C (Fig. 2a). At relatively low temperature of 240 °C and low pressure of 50 bar with a CO conversion of 17%, a high C₂₋₄OH selectivity of 45.2% can be achieved over the catalyst, which is higher than those of

previously reported non-noble metal-based catalysts under similar conditions (Fig. 2c, Supplementary Table 3). Moreover, a high yield of 8.41% for C₂₋₄OH can also be achieved at 240 °C and 50 bar, while similar yield can only be reached at temperatures above 300 °C and pressures above 80 bar over previously reported MoS₂-based catalysts (Supplementary Fig. 5, Supplementary Table 4). Interestingly, the distribution of carbon number in the alcohol products is quite narrow and the selectivity of C₂₋₄OH in C₂₊OH is almost 100%, indicating the well-controlled carbon-chain growth over the ER-MoS₂-K catalyst compared with classical Co-based and Fe-based catalysts[12,28]. In contrast to the ER-MoS₂-K catalyst, methanol becomes the main alcohol product over the PD-MoS₂-K catalyst with abundant basal planes, especially at lower reaction temperatures (Fig. 2b), and the C₂₋₄OH selectivity can only reach 24.5% at best at 240 °C. The formation rate of C₂₋₄OH (calculated on carbon mol basis) over the PD-MoS₂-K catalyst is only 0.96 mmol $g_{cat.}^{-1}$ $h^{-1}$ compared with that of 2.0 mmol $g_{cat.}^{-1}$ $h^{-1}$ over the ER-MoS₂-K catalyst. Consequently, the C₂₋₄OH to methanol selectivity ratio is overturned from 0.4 to 2.2 by reducing the lateral size of MoS₂ to enrich the edges (Fig. 2d), thus indicating the important role of MoS₂ edges in promoting the CO hydrogenation to C₂₋₄OH.

The ER-MoS₂-K catalyst exhibits a high stability with well-maintained CO conversion and C₂₋₄OH selectivity in the performance test for 100 h (Fig. 2e). TEM and X-ray absorption spectroscopy (XAS) characterizations show no obvious change in the structure of the ER-MoS₂-K catalyst after the 100 h of on-stream reaction test (Supplementary Fig. 3a, b, and d). To further investigate the long-term durability of the ER-MoS₂-K catalyst, the catalytic performances for CO hydrogenation reaction with time-on-stream (TOS) of ~1000 h were also evaluated. Despite desulfurization of MoS₂ as a common problem causing activity loss is unavoidable[31], the catalytic performance of ER-MoS₂-K can be sustained for TOS of 1075 h via regularly replenishing sulfur at every TOS of 300 h by using H₂S/H₂ treatment of the catalyst (Supplementary Fig. 6). XRD and HAADF-STEM characterizations show the well-maintained 2H-MoS₂ crystal phase and channel structure of the used ER-MoS₂-K catalyst after the long-term reaction test

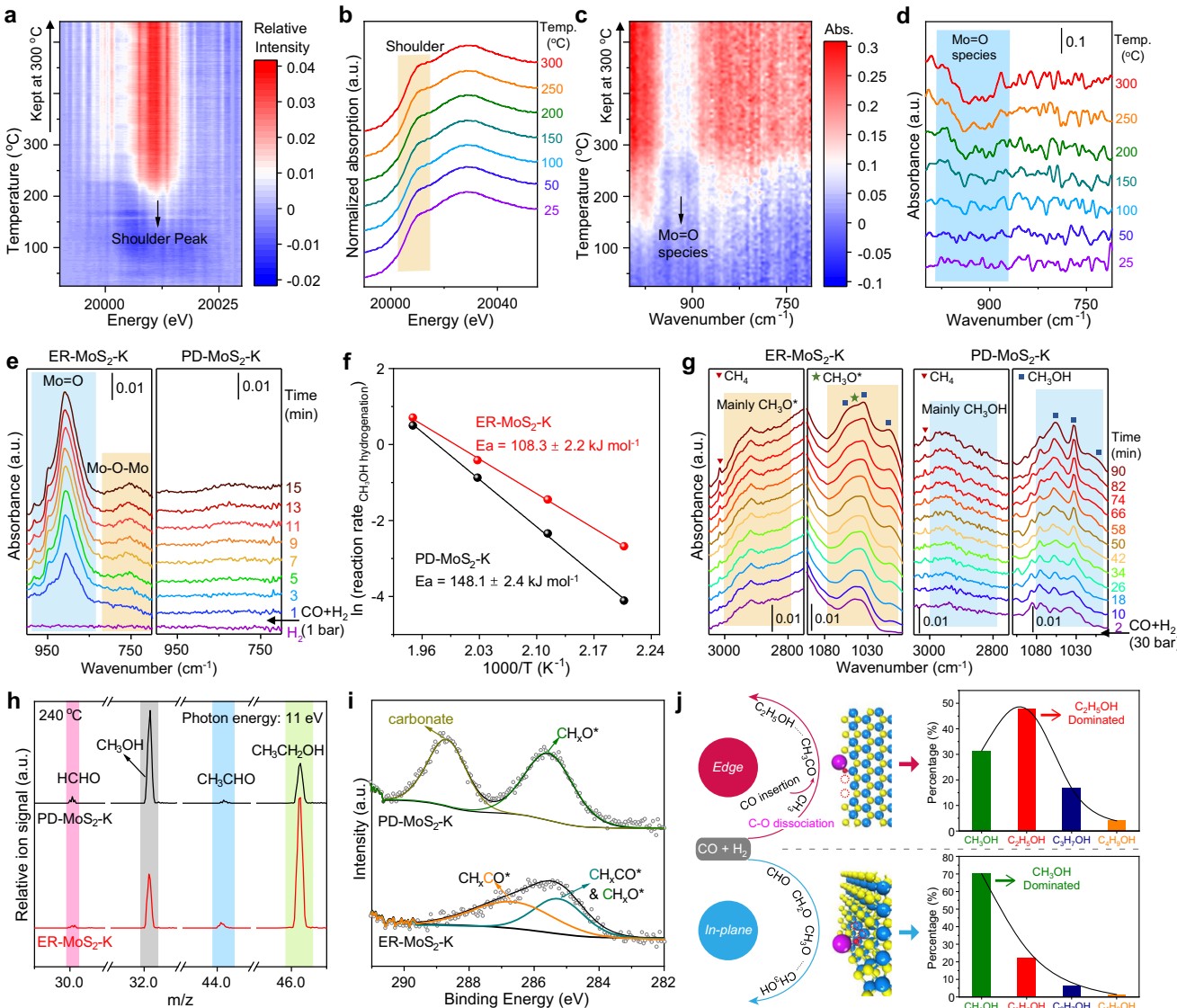

**Fig. 3 | C-O dissociation ability and CO hydrogenation mechanism of different MoS₂ catalysts. a–d** In-situ ED-XAS (**a**, **b**) coupling in-situ DRIFTS (**c**, **d**) characterizations of ER-MoS₂-K during H₂ pretreatment measured with a hyphenated technology. Relative intensity in (**a**) was obtained by subtracting each spectrum to the first spectrum. **e** In-situ DRIFT spectra of CO hydrogenation over the H₂-pretreated catalysts at 25 °C, H₂/CO of 2 and 1 bar. **f** Arrhenius plots calculated based on the reaction rate for methanol hydrogenation over PD-MoS₂-K and ER-MoS₂-K. **g** In-situ DRIFT spectra of CO hydrogenation over the H₂-pretreated catalysts at 240 °C, H₂/CO of 2 and 30 bar. **h** In-situ SVUV-PIMS detection of the CO hydrogenation intermediates and products over the H₂-pretreated catalysts at 240 °C, H₂/CO of 2 and 5 bar. **i** Quasi in-situ SRPES detection of the CO hydrogenation intermediates. SRPES spectra were obtained after the treatment of the H₂-treated catalysts with reaction gas (H₂/CO = 2) at 240 °C and 5 bar for 0.5 h. **j** Proposed reaction mechanism for CO hydrogenation to methanol and ethanol on edge and in-plane sites of MoS₂.

(Supplementary Figs. 6c and 7). These results reveal the ER-MoS₂-K as a potential catalyst for industrial application combined with regular sulfur maintenance.

To exclude the influence of different morphologies of the ER-MoS₂-K and PD-MoS₂-K on their catalytic performances and also to further verify the effect of MoS₂ edge in improving the selectivity of C₂₋₄OH, we prepared another two K-promoted MoS₂ catalysts possessing similar foam-like morphologies, but consisting of MoS₂ nanosheets with different structural features, i.e. edge-rich nanosheets and basal plane-dominated nanosheets, which are denoted as ER-MoS₂ᶠᵒᵃᵐ-K and PD-MoS₂ᶠᵒᵃᵐ-K, respectively. These two catalysts were synthesized by using SiO₂ sphere as a hard template but with different sulfur precursors (see Methods part for detailed synthesis process, Supplementary Fig. 8). XRD patterns demonstrate that the foam-like MoS₂ samples exhibit typical characteristics of hexagonal 2H-MoS₂ crystal (Supplementary Fig. 8i). The almost identical Mo K-edge XAS

spectra and XPS spectra of different MoS₂ samples demonstrate that the chemical states of Mo and S in the foam-like samples (ER-MoS₂ᶠᵒᵃᵐ and PD-MoS₂ᶠᵒᵃᵐ) are basically the same with those in the ER-MoS₂ and PD-MoS₂ (Supplementary Fig. 9). In addition, NH₃ temperature-programmed desorption (NH₃-TPD) experiment[32] of all catalysts show similar desorption peaks at ~130 °C corresponding to NH₃ desorption from the coordinatively unsaturated Mo sites at sulfur vacancies as weak Lewis acid sites (Supplementary Fig. 10), suggesting that the effect of acidity of different catalysts on their performances is neglectable. These results indicate that the foam-like catalysts are reliable references to exclude the morphology effect.

Structural characterizations confirm that both ER-MoS₂ᶠᵒᵃᵐ-K and PD-MoS₂ᶠᵒᵃᵐ-K possess uniform porous frameworks with similar surface areas and pore sizes (Supplementary Fig. 8a, b, d, e, and Supplementary Table 1, Entry 8, 10), yet are different in the lateral size of the MoS₂ nanosheets (Supplementary Fig. 8c, f). The ER-MoS₂ᶠᵒᵃᵐ-K is

featured by small-sized nanosheets with abundant edges (Supplementary Fig. 8e, f), while the PD-MoS$_2^{foam}$-K mainly exposes large basal planes of MoS$_2$ as shown in the TEM images (Supplementary Fig. 8b, c). The edge-length statistical analysis based on the TEM images also demonstrates the markedly smaller lateral size of the ER-MoS$_2^{foam}$-K than that of the PD-MoS$_2^{foam}$-K (9 nm $vs.$ 741 nm, Supplementary Fig. 8g, h). Interestingly, reaction performance tests show that an obviously higher C$_{2-4}$OH to methanol selectivity ratio of 1.4 is obtained over the ER-MoS$_2^{foam}$-K compared with that of 0.6 over the PD-MoS$_2^{foam}$-K (Supplementary Fig. 8j, Supplementary Table 5), which is consistent with the comparison between the ER-MoS$_2$-K and PD-MoS$_2$-K (Fig. 2d, Supplementary Table 5). These results further confirm the key role of MoS$_2$ edges in catalyzing CO hydrogenation to C$_{2-4}$OH.

## Mechanistic understanding of higher alcohols synthesis over MoS$_2$

To further understand the catalytic function of MoS$_2$ edges in CO hydrogenation, the distribution ratio of CH$_x$ and CH$_x$O parts (n$_{CHx}$/n$_{CHxO}$) in the non-CO$_2$ products is analyzed and used as an indicator for the C-O bond dissociation activity of the MoS$_2$ catalysts (Supplementary Fig. 11), since CH$_x$ comes from the dissociated CH$_x$O. For CO hydrogenation over the ER-MoS$_2$-K at 240 °C, the obtained hydrogenated products present a n$_{CHx}$/n$_{CHxO}$ of 1.45, which is remarkably higher than that of 0.44 over the basal plane-dominated PD-MoS$_2$-K catalyst. These results show that C-O cleavage is more favorable over the ER-MoS$_2$-K compared with that over the PD-MoS$_2$-K, thus indicating that the edge sites of MoS$_2$ can facilitate the C-O dissociation of CH$_x$O to produce CH$_x$ intermediate.

To gain deep insights into the active sites at the MoS$_2$ edges, the dynamic change in electronic states and evolution of surface oxygen species during the H$_2$ reduction pretreatment and subsequent reaction test of the catalyst were monitored by using a hyphenated technology of in-situ time-resolved energy-dispersive X-ray absorption spectroscopy (ED-XAS) and in-situ diffuse reflectance infrared Fourier transform spectroscopy (DRIFTS) characterizations (Supplementary Fig. 12a). The in-situ ED-XAS results show that the absorption edge gradually shifts to higher energies together with an increase in the intensity of its shoulder peak at around 20010 eV during the H$_2$ pretreatment process of both ER-MoS$_2$-K and PD-MoS$_2$-K catalysts, indicating the decrease of the Mo oxidation state with the exposure of coordinatively unsaturated Mo atoms (Fig. 3a, b and Supplementary Fig. 12b, c)[30,33], which can be ascribed to the formation of sulfur vacancies due to the removal of surface O atoms and some S atoms from MoS$_2$ by the H$_2$ pretreatment. This is also reflected by the directly detected signals of H$_2$O, H$_2$S, and SO$_2$ during the reduction process in the in-situ mass spectrum for both catalysts (Supplementary Fig. 13). Electron paramagnetic resonance (EPR) characterizations show the rise of a signal at $g$ = 2.00, further confirming the existence of SVs on both the reduced ER-MoS$_2$-K and PD-MoS$_2$-K catalysts (Supplementary Fig. 14a)[34,35]. Moreover, both catalysts possess similar densities of SVs as reflected by their similar integrated intensities of EPR peaks (Supplementary Fig. 14b) and also similar adsorption capacities of NO (89.1 $vs.$ 103.3 umol$_{NO}$ g$_{cat.}^{-1}$) (Supplementary Fig. 14c). Performance of ER-MoS$_2$-K is substantially improved after the H$_2$ pretreatment, suggesting the key role of SVs in catalyzing CO hydrogenation (Supplementary Fig. 15).

The removal of surface oxygen is also evidenced by in-situ DRIFTS characterizations (hyphenated with in-situ ED-XAS) of the process, displaying the decreased intensity of the absorption peaks for the vibrations of Mo=O and Mo-O-Mo at around 930 cm$^{-1}$ and 760 cm$^{-1}$, respectively (Fig. 3c, d and Supplementary Fig. 12d, e), which can be attributed to the removal of oxygen adsorbed at the edge and in-plane SVs, respectively[30,36]. It's worth noting that only the removal of Mo=O species was obviously observed during H$_2$ pretreatment of the ER-MoS$_2$-K catalyst (Fig. 3c, d), suggesting that the ER-MoS$_2$-K catalyst is

enriched with edge SVs. In contrast, the PD-MoS$_2$-K catalyst possesses more in-plane SVs as reflected by the removal of mainly Mo-O-Mo species during H$_2$ pretreatment (Supplementary Fig. 12d, e). In-situ O$_2$-adsorption DRIFTS characterizations of the K-free MoS$_2$ samples show the rise of mainly Mo=O species on ER-MoS$_2$ while mainly Mo-O-Mo species on PD-MoS$_2$, also suggesting the totally different SVs distributions on the two kinds of catalysts (Supplementary Fig. 16).

After the in-situ H$_2$ pretreatment of the catalysts for the generation of SVs, reaction tests were then performed by using CO/H$_2$ mixture or Ar-bubbled CH$_3$OH as the feed gas, respectively. In both cases, in-situ DRIFTS characterizations clearly show the reappearance of the Mo-O peaks over the H$_2$-pretreated ER-MoS$_2$-K catalyst (Fig. 3e, Supplementary Fig. 17), which can be attributed to the formation of Mo-O species from the C-O dissociation of CH$_x$O at the SVs sites. In comparison, only negligible Mo-O peaks with much weaker intensities were detected over the H$_2$-pretreated PD-MoS$_2$-K catalyst either with syngas or methanol as the feed gas (Fig. 3e and Supplementary Fig. 17). In addition, the hydrogenation of CH$_3$OH to methane reaction was also tested over the two catalysts, in which the ER-MoS$_2$-K exhibits a higher activity with a lower activation energy compared with the PD-MoS$_2$-K based on the Arrhenius plots (Fig. 3f). These results demonstrate that the SVs at the MoS$_2$ edge possess a higher activity for the C-O dissociation than those in the basal plane.

To obtain mechanistic understanding of the CO hydrogenation reaction process on MoS$_2$-based catalysts, a series of in-situ characterizations were conducted to capture the reaction intermediates. In-situ high-pressure DRIFTS characterizations show the gradual appearance of surface CH$_3$O* species and CH$_3$OH on both ER-MoS$_2$-K and PD-MoS$_2$-K when CO/H$_2$ mixture passes through the catalysts at 240 °C (Fig. 3g), where the vibrational bands at 2957, 2916, 2855 and 1044 cm$^{-1}$ can be attributed to CH$_3$O* species, while the complicated vibrational bands from 2800 ~ 3000 cm$^{-1}$ and bands at 1056, 1032, 1012 cm$^{-1}$ correspond to the C-H and C-O stretch of CH$_3$OH, respectively[37]. It is noteworthy that formation of CH$_3$OH was observed as dominating species on the PD-MoS$_2$-K catalyst in contrast to the dominating CH$_3$O* on the ER-MoS$_2$-K (Fig. 3g)[38]. This is supported by our density functional theory (DFT) calculations that the hydrogenation of CH$_3$O* to CH$_3$OH* is exergonic by 0.81 eV with a low reaction barrier of 0.38 eV over the in-plane SVs but is endergonic by 0.6 eV with a higher barrier of 0.96 eV over the Mo-edge SVs (Supplementary Fig. 18), thus indicating the more favorable formation of CH$_3$OH on the PD-MoS$_2$-K with rich basal planes than that on the ER-MoS$_2$-K with rich edges under the same reaction condition (Fig. 3g). These results explain the higher selectivity toward CH$_3$OH in CO hydrogenation over the PD-MoS$_2$-K catalyst. As the H$_2$-pretreated catalysts were exposed to 5 bar of reaction gas (H$_2$/CO = 2) at 240 °C, signals of HCHO, CH$_3$OH, CH$_3$CHO, and CH$_3$CH$_2$OH were directly detected by using in-situ synchrotron-based VUV photoionization mass spectrometry (SVUV-PIMS) (Fig. 3h). HCHO and CH$_3$CHO could be reaction intermediates for the generation of CH$_3$OH and CH$_3$CH$_2$OH. The intensity ratio of CH$_3$OH/CH$_3$CH$_2$OH follows the trend as mentioned above (Fig. 2a, b) that the main product is CH$_3$OH for the PD-MoS$_2$-K catalyst and CH$_3$CH$_2$OH for the ER-MoS$_2$-K catalyst.

Quasi in-situ synchrotron radiation photoelectron spectroscopy (SRPES) measurements were conducted to further detect the surface adsorbed species on catalysts (Fig. 3i). Before the measurements, the reduced ER-MoS$_2$-K and PD-MoS$_2$-K catalysts were treated with the reaction gas (H$_2$/CO = 2/1, 5 bar) at 240 °C for 0.5 h. At the binding energy region of C 1s, two peaks at 285.3 eV and 286.7 eV were detected on the ER-MoS$_2$-K catalyst, which can be assigned to the CH$_x$O* and CH$_x$CO* species[39]. In contrast, only CH$_x$O* and carbonate species corresponding to the peaks at 285.6 eV and 288.8 eV were detected without C-C coupled CH$_x$CO* species on the PD-MoS$_2$-K catalyst. Based on the above results, we propose that the hydrogenation of CO to CH$_3$CH$_2$OH undergoes the following steps,

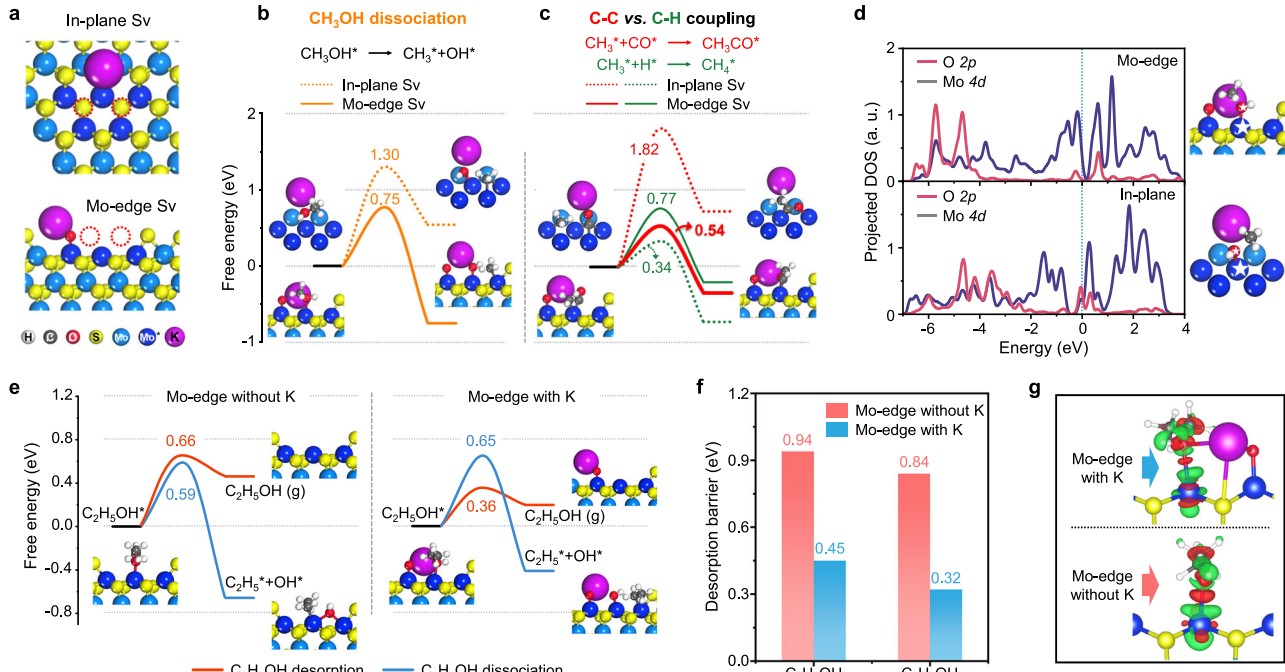

**Fig. 4 | DFT studies of C-O cleavage, C-C coupling, and the desorption of alcohols on double-Sv of MoS₂. a** Models of potassium-modified in-plane SVs and Mo-edge SVs. **b** DFT calculated reaction barrier for the dissociation of CH₃OH on Mo-edge SVs and in-plane SVs. **c** Reaction barrier of the hydrogenation of CH₃* and the insertion of CO into CH₃* on Mo-edge SVs and in-plane SVs. **d** Projected DOS of Mo 4d orbits and O 2p orbits of transition states of methanol dissociation on Mo-edge SVs (up) and in-plane SVs (down). **e** DFT calculations on C₂H₅OH* desorption or dissociation on Mo-edge SVs without or with the decoration of potassium. **f** DFT calculated reaction barriers for the desorption of C₃H₇OH and C₄H₉OH on Mo-edge SVs with or without the decoration of potassium. **g** Side view of differential charge density of the adsorbed C₂H₅OH on Mo atoms in Mo-edge SVs with or without the decoration of potassium. The red and green regions indicate electron accumulation and depletion, respectively.

i. the adsorption and hydrogenation of CO to CH₃O* (or CH₃OH*),
ii. the dissociation of CH₃O* (or CH₃OH*) to CH₃*,
iii. the insertion of CO* into CH₃* to produce CH₃CO*,
iv. the hydrogenation of CH₃CO* to CH₃CHO* and then CH₃CH₂OH*,
v. the desorption of CH₃CH₂OH*.

The edge SVs of MoS₂ favor more the synthesis of C₂₊OH by facilitating the dissociation of C-O bond of CHₓO* and then the carbon-chain growth via C-C coupling (Fig. 3j).

**Key factors for tailoring carbon-chain growth**
The C-O dissociation and C-C coupling are the key steps for controlling the carbon-chain growth in CO hydrogenation. To gain an atomic level understanding of the higher C₂₊OH selectivity over the edge SVs than that over the in-plane SVs, the activity of the two types of SVs for carbon-chain growth were studied by using DFT calculations. The dissociation of CH₃OH* and coupling of CₓH₂ₓ₊₁* (x = 1–3) with CO* or H* were adopted as probing reactions for investigating the C-O dissociation and C-C coupling activities, since these intermediates are common surface species during the reaction. A double-Sv at the Mo-edge or in the basal plane was simulated as the active site (Supplementary Fig. 19a, d). The potassium promoter was introduced into the model by forming a Mo-O-K structure in the adjacent of the double-Sv (Fig. 4a and Supplementary Fig. 19b, c, e, f). This is based on in-situ DRIFTS characterizations of the H₂ reduction process, which show that the Mo=O species of the ER-MoS₂ can almost be completely removed by H₂, while a certain amount of Mo=O species of the ER-MoS₂-K remains stable even at 300 °C under the decoration of potassium (Supplementary Fig. 20), indicating the formation of Mo-O-K structure with a higher stability than the normal Mo=O.

The dissociation of CH₃OH* to CH₃* and OH* over the Mo-edge SVs requires an activation energy of only 0.75 eV, which is notably lower than that of 1.30 eV over the in-plane SVs (Fig. 4b). This indicates that the Mo-edge SVs possess a higher activity for the dissociation of C-O bond to form CH₃ species, which is essential for the subsequent C-C coupling steps. Analyses of projected density of states (PDOS) reveal the formation of a stronger Mo-O bond in the transition state for CH₃OH dissociation over the Mo-edge SVs compared with that over the in-plane SVs, as indicated by the lower energy levels of the over-lapped part of Mo 4d and O 2p orbits in the transition states of the former case (Fig. 4d). This leads to lowered C-O dissociation barrier on the Mo-edge SVs.

The reaction energetics of the CₓH₂ₓ₊₁*-CO* (C-C) coupling and CₓH₂ₓ₊₁*-H* (C-H) coupling was then investigated on the in-plane and Mo-edge SVs. On the in-plane SVs, the CH₃*-CO* coupling requires a much higher activation energy of 1.82 eV than the CH₃*-H* coupling with only a low barrier of 0.34 eV (Fig. 4c). However, on the Mo-edge SVs, the barrier of CH₃*-CO* coupling (0.54 eV) becomes notably lower than that of CH₃*-H* coupling (0.77 eV), and similar results was also found in the cases of C₂H₅* (0.42 eV for C-C coupling vs 0.76 eV for C-H coupling) and C₃H₇* (0.53 eV for C-C coupling vs 0.81 eV for C-H coupling) (Supplementary Fig. 21). In addition, the formation of CₓH₂ₓ₊₁CO* from C-C coupling on the Mo-edge SVs is also more exergonic in free energy than the formation of CₓH₂ₓ₊₂ from C-H coupling (Supplementary Fig. 21). These results demonstrate that the C-C coupling between CₓH₂ₓ₊₁* and CO* for carbon-chain growth is thermodynamically and kinetically more favorable than the C-H coupling between CₓH₂ₓ₊₁* and H* for alkane formation on the edge SVs of MoS₂.

Potassium modification plays a key role in improving the selectivity toward alcohols in CO hydrogenation over MoS₂-based catalysts (Supplementary Fig. 22 and 23). With the decoration of potassium, the selectivity of alcohols significantly increases from <1.5% over the ER-MoS₂ catalyst to >60% over the ER-MoS₂-K catalyst at 240 °C (Supplementary Fig. 22a). The above analysis shows that the Mo-edge SVs

are highly active for the dissociation of C-O bond, which may lead to decomposition of the formed alcohols. Thus, the timely desorption of the formed alcohols to avoid undesired C-O cleavage is critical for the production of alcohols. To understand the effect of potassium in promoting the production of alcohols, we further conducted DFT calculations for the desorption and C-O cleavage of $CH_3OH$ and $C_2H_5OH$ on Mo-edge SVs with or without the decoration of potassium (Fig. 4e and Supplementary Fig. 24). In the case without the modification of potassium, the C-O cleavage of $CH_3OH^*$ and $C_2H_5OH^*$ requires activation energies of 0.73 eV and 0.59 eV, respectively, which are lower than the activation energies of 0.95 eV and 0.66 eV for the competing desorption of $CH_3OH^*$ and $C_2H_5OH^*$, respectively. In contrast, after decorating the Mo-edge SVs with potassium, the activation energies for $CH_3OH^*$ and $C_2H_5OH^*$ desorption are significantly decreased to 0.45 eV and 0.36 eV, respectively, which are markedly lower than the activation energies of 0.75 eV and 0.65 eV for C-O cleavage of $CH_3OH^*$ and $C_2H_5OH^*$, respectively. In addition, the difference between the free energies of desorption and dissociation states is also notably decreased after potassium decoration, making the desorption more favorable while dissociation less favorable in thermodynamics compared with those without potassium. Similar effects were also observed in the desorption of $C_3H_7OH$ and $C_4H_9OH$ (Fig. 4f). These results suggest that potassium modification has little effect on the activation energies for C-O cleavage as also evidenced by the similar apparent activation energies for the hydrogenation of $CH_3OH$ to $CH_4$ and $C_2H_5OH$ to $C_2H_6$ on the ER-$MoS_2$ and ER-$MoS_2$-K (Supplementary Fig. 25b, c), but has a significant effect in promoting the kinetics and thermodynamics for alcohols desorption, which contributes to the hindered deep-hydrogenation of alcohols (Supplementary Fig. 25) and thus the enhancement of alcohols selectivity.

The effect of potassium in promoting the formation and desorption of alcohols is also verified by the in-situ high pressure DRIFTS characterizations for the CO hydrogenation over the ER-$MoS_2$ and ER-$MoS_2$-K, in which the gradually enhanced vibration bands of alcohols can be observed on the ER-$MoS_2$-K at elevated temperatures, whereas only the alkyl species was detected on the ER-$MoS_2$ surface under identical reaction conditions (Supplementary Fig. 26). The facilitated desorption of alcohols on potassium modified Mo-edge SVs can be attributed to the electrostatic interaction between the potassium and the hydroxyl of alcohols, which weakens their adsorption on the Mo site, as reflected by the decreased charge density in the Mo-O bonding region when potassium exists (Fig. 4g). These results support that the potassium-decorated edge SVs of $MoS_2$ are the active sites for the selective CO hydrogenation to higher alcohols.

## Discussion

The precise regulations of the activities for C-O cleavage and C-C coupling are always the challenging tasks in selective CO hydrogenation to specific $C_{2+}OH$. In this study, we reveal the superiority of $MoS_2$-edge SVs in facilitating simultaneously the C-O cleavage and C-C coupling, and the key role of potassium in promoting the timely desorption of alcohols via electrostatic interaction with hydroxyls of alcohols, thereby enabling the controlled carbon-chain growth for the preferential formation of $C_{2-4}OH$. Highly selective CO hydrogenation to $C_{2-4}OH$ was achieved over potassium-modified edge-rich $MoS_2$, which exhibits high CO conversion of 17% and $C_{2-4}OH$ selectivity of 45.2% in hydrogenated products at 240 °C and 50 bar, surpassing previously reported non-noble metal-based catalysts under similar conditions. In contrast, methanol was always the primary product at a wide range of reaction temperatures over the basal plane-dominated $MoS_2$-based catalyst. This work presents the high flexibility of edge SVs of $MoS_2$ in tailoring both C-O cleavage and C-C coupling for carbon-chain growth in CO hydrogenation, thus providing a prototype for the rational design of the nanostructure and microenvironment of active sites for selective hydrogenation reactions.

## Methods

### Preparation of catalysts

The Edge-rich $MoS_2$ (ER-$MoS_2$) was synthesized by using the SBA−15 as template. Typically, 400 mg $(NH_4)_6Mo_7O_{24}\cdot4H_2O$ and 2.5 g SBA-15 were dispersed in 50 mL deionized water and followed by drying at room temperature under magnetic stirring. The obtained power was then dried overnight under 80 °C. The dried product and 10 mL $CS_2$ were sealed into a 40 mL stainless steel autoclave under Ar protection and maintained at 400 °C for 4 h. After that, the HF solution treatment under room temperature for >8 h was conducted to remove SBA-15 template. The final product was then washed with water and absolute ethanol for several times and was dried at 80 °C.

The PD-$MoS_2$ was synthesized by using the method of our previous work[40]. Typically, 900 mg $(NH_4)_6Mo_7O_{24}\cdot4H_2O$ was dissolved in 20 mL deionized water to form a homogeneous solution. Then, the solution and 10 mL $CS_2$ were sealed into a 40 mL stainless steel autoclave under Ar protection and maintained at 400 °C for 4 h. The product was treated with 6 mol $L^{-1}$ KOH solution under stirring at 60 °C for 3 h, followed by washing with pure water and absolute ethanol for several times and then drying at 80 °C.

The ER-$MoS_2^{foam}$ and the PD-$MoS_2^{foam}$ were synthesized by using the $SiO_2$ sphere as template. For the synthesis of $SiO_2$ sphere, 300 mL ethanol, 100 mL deionized water and 28 mL $NH_3\cdot H_2O$ (aq.) were mixed. After 10 min of stir, 26 mL tetraethyl orthosilicate (TEOS) was added into the solution, followed by sealed drastic stirring for 5 h to form $SiO_2$ sphere. The obtained $SiO_2$ sphere was collected by centrifugation, and was washed by water and absolute ethanol for several times. After that, the product was dried overnight at 80 °C. For the synthesis of ER-$MoS_2^{foam}$, 400 mg $(NH_4)_6Mo_7O_{24}\cdot4H_2O$ and 2.5 g $SiO_2$ sphere were dispersed in 50 mL deionized water and followed by drying at room temperature under magnetic stirring. The obtained power followed the same reaction and treatment process as that of ER-$MoS_2$. For the synthesis of PD-$MoS_2^{foam}$, 400 mg $(NH_4)_6Mo_7O_{24}\cdot4H_2O$, 800 mg thiourea and 2.5 g $SiO_2$ sphere were dispersed in 50 mL deionized water and followed by drying at room temperature under magnetic stirring. The obtained power was then dried overnight under 80 °C. The dried product was sealed into a 40 mL stainless steel autoclave under Ar protection and maintained at 400 °C for 4 h. After that, the product followed the same treatment process as that of ER-$MoS_2$.

$MoS_2$ materials were decorated by potassium before catalytic test, which was used as a promoter without affecting the morphology and crystal form of $MoS_2$. Typically, $MoS_2$ was dispersed in 20 mL deionized water by ultraphonic for 30 min. Then, the $K_2CO_3$ (a.q.) was dispersed into the dispersion by ultraphonic for another 30 min, followed by drying at room temperature under magnetic stirring. The obtained product was then dried under 80 °C. The mole ratio of K/Mo was controlled in 0.2 unless otherwise stated.

### Evaluation of catalytic performance

The catalytic reactions were performed with a high-pressure fixed-bed reactor equipped with gas chromatograph (GC). Typically, before the reaction, 200 mg catalyst in grain sizes of 250-600 μm was pretreated in-situ in a $H_2$ gas flow of 30 mL $min^{-1}$ at 1 bar and a temperature of 300 °C for 1 - 3 h. After the reduction, the reactant was introduced into the reactor. Ar was mixed into the $H_2$/CO mixture as an internal standard for the calculation of CO conversion. The reactions were operated under a pressure of 50 bar and temperatures range from 200 °C to 280 °C, with a $H_2$/CO ratio of 2/1.

The methanol and ethanol hydrogenation experiments were also performed in high-pressure fixed-bed reactor with 100 mg catalyst. After the pre-reduction process as described above, liquid methanol or ethanol (0.005 mL $min^{-1}$) was fed into the reactor using a liquid pump, meanwhile, high purity $H_2$ was fed into the reactor at a gas flow rate of 30 mL $min^{-1}$ at 1 bar and temperatures from 180 °C to 240 °C.

Products were analyzed by an online gas chromatograph, which was equipped with a thermal conductivity detector (TCD) and a flame ionization detector (FID). TDX-01 packed column was connected to TCD, and RT-Q-BOND-PLOT capillary column was connected to FID. The product selectivity was calculated on a molar carbon basis. The catalytic performances after 20 h of reaction were typically used for discussion.

The forward reaction rate ($r_f$) was calculated on the basis of the measured net reaction rate ($r_n$) and equilibrium factor ($\eta$):[41]

$$r_f = r_n/(1 - \eta) \quad (1)$$

As an example, for the reaction CO (g) + 2H$_2$ (g) $\rightleftharpoons$ CH$_3$OH (g), $\eta$ was calculated using

$$\eta = \frac{P_{CH_3OH}}{P_{CO} \times P_{H_2}^2} \times \frac{1}{K} \quad (2)$$

Where $P_x$ is the partial pressure of species $x$ (in atm) and $K$ is the equilibrium constant for the reaction, which is $3.271 \times 10^{-3}$ atm$^{-2}$ at 240 °C. The calculated forward reaction rates in this work are almost identical to the net reaction rates.

## Catalyst characterization

Transmission electron microscopy (TEM), High-angle annular dark field scanning transmission electron microscopy (HAADF-STEM) and energy-dispersive X-ray (EDX) mapping measurements were carried out on a Phillips Analytical FEI Tecnai20 electron microscope or on a JEOL ARM-200F field-emission transmission electron microscope operated at an acceleration voltage of 200 kV. Scanning electron microscopy (SEM) measurements were carried out using a Hitachi S-4800 scanning electron microscope with a 15 kV accelerating voltage.

X-ray fluorescence (XRF) characterizations were conducted on a Zetium XRF spectrometer. Prior to measurement, the sample was pretreated with H$_2$ at 300 °C for 1 h and subsequently purged with Ar at 25 °C overnight. Inductively coupled plasma optical emission spectrometry (ICP-OES) was conducted on Shimadzu ICPS-8100. The C and S contents in the used catalysts were analyzed using a EMIA-8100 elemental analyzer.

X-ray diffraction (XRD) patterns were recorded on a Rigaku Ultima IV diffractometer with Cu Kα radiation (λ = 0.15406 nm) at 40 kV and 30 mA. The diffraction angles were scanned from 5 to 85 degrees (2θ) with a speed of 10-degree min$^{-1}$.

X-ray absorption fine structure (XAFS) spectra were measured at the BL14W1 beamline of the Shanghai Synchrotron Radiation Facility (SSRF).

N$_2$ adsorption measurements were performed on Quadrasorb evo. Prior to N$_2$ adsorption, the samples were degassed under vacuum at 120 °C for 6 h.

NH$_3$ temperature-programmed desorption (NH$_3$-TPD) and NO chemisorption measurements were performed on a Micromeritics Auto Chem II 2920 instrument. Prior to each measurement, the sample was pretreated in situ with H$_2$ at 300 °C for 3 h and subsequently purged with He at 300 °C for 3 h. For NH$_3$-TPD, the adsorption of NH$_3$ was performed at 50 °C in He gas containing 10% NH$_3$ for 1 h and TPD was performed in He flow by raising the temperature to 800 °C with a rate of 10 °C min$^{-1}$. The NO chemisorption experiments were performed at 40 °C by pulsing 0.5 mL of 2% NO/He (0.314 μmol NO per pulse) through the catalyst every 30 min[30,42]. The NO effluent (detected by mass spectrometer) gradually increased to a constant value, signifying NO saturation of the catalyst, and then the total uptake of NO was calculated.

X-ray photoelectron spectroscopy (XPS) characterizations were performed on ThermoFisher ESCALAB 250Xi spectrometer or SPECS spectrometer using Al Kα x-rays as the excitation source. XPS characterizations of the used catalysts after ~1000 h of on-stream reaction were conducted without exposing the samples to air.

Characterizations using the hyphenated technology of in-situ time-resolved energy dispersive-X-ray absorption spectroscopy (ED-XAS) and in-situ diffuse reflectance infrared Fourier transform spectroscopy (DRIFTS) were performed at D-line (BL05U&BL06B1) of Shanghai Synchrotron Radiation Facility. Before measurement, MoS$_2$ catalyst powder was loaded into the cell of in-situ reactor. Then, a mixture gas of 36%H$_2$/Ar was introduced into the reactor. After that, the reactor was heated to 300 °C with a rate of 5 °C min$^{-1}$, and was held at 300 °C for 1 h. Spectra were recorded during the whole in-situ experiment.

Quasi in-situ electron paramagnetic resonance (EPR) spectroscopic measurements were performed on a Bruker A200 EPR spectrometer operated at X-band frequency with a microwave frequency of 9.32 GHz, a microwave power of 10 mW and a modulation frequency of 100 kHz. EPR characterization was conducted using a quasi in-situ method, in which the sample was reduced in the sample tube, and then the sample tube was transferred in EPR spectrometer for measurement. Sample was not exposed to air during the whole process. Typically, catalyst powder was placed in a quartz EPR sample tube. Before measurement, the catalyst was pretreated by H$_2$ for 3 h at 300 °C, and then was purged with Ar at 300 °C for 1 h. All spectra were normalized with the mass of catalysts.

In-situ diffuse reflectance infrared Fourier transform spectroscopy (DRIFTS) measurements were performed using a Fourier transform infrared spectrometer (Nicolet 6700 for atmospheric pressure test and vertex 70 V for high pressure test) equipped with a mercury cadmium telluride detector. The in-situ DRIFT spectra were recorded by collecting 64 scans at a resolution of 4 cm$^{-1}$. Before measurement, all catalysts were pretreated in-situ in a H$_2$ gas flow of 30 mL min$^{-1}$ at 1 bar and a temperature of 300 °C for 1 h. For in-situ DRIFTS measurement of CO hydrogenation at atmospheric pressure, catalysts were subsequently cooled down to 25 °C in H$_2$ flow after H$_2$ pretreatment, then the background spectra were obtained. After that, catalysts were exposed to syngas (CO/H$_2$ = 1/2) in a gas flow of 30 mL min$^{-1}$ with a pressure of 1 bar. For in-situ DRIFTS measurement of methanol adsorption at atmospheric pressure, after H$_2$ pretreatment, catalysts were purged with Ar for 30 min and subsequently cooled down to 25 °C in Ar flow, then the background spectra were obtained. After that, catalysts were exposed to methanol steam, which was introduced by Ar. For in-situ DRIFTS measurement of CO hydrogenation at high pressure, catalysts were subsequently cooled down to reaction temperature in H$_2$ flow after H$_2$ pretreatment, then the background spectra were obtained. After that, catalysts were exposed to syngas (CO/H$_2$ = 1/2) in a gas flow of 30 mL min$^{-1}$ with a pressure of 30 bar. Spectra were recorded during the whole in-situ DRIFTS experiment.

In-situ Synchrotron-based VUV photoionization mass spectrometry (SVUV-PIMS) study was carried out at the mass spectrometry end-station of the National Synchrotron Radiation Laboratory at Hefei, China[43,44]. Before experiment, 200 mg catalysts were treated in-situ by H$_2$ at 300 °C for 1 h. To detect the intermediates and products during CO hydrogenation, the reduced catalysts were exposed to CO/H$_2$ (1/2) atmosphere with a flow rate of 50 mL min$^{-1}$ and a pressure of 5 bar at 240 °C, the photoionization mass spectra were collected for 300 s at a photon energy of 11 eV.

Quasi in-situ synchrotron radiation photoelectron spectroscopy (SRPES) measurements were performed at the photoemission end-station at the beamline BL10B of the National Synchrotron Radiation Laboratory (NSRL) in Hefei, China. The beamline is connected to a bending magnet and covers photon energies from 100 to 1000 eV with a resolving power ($E/\Delta E$) better than 1000. The end-station is composed of four chambers: an analysis chamber, a preparation chamber,

a load-lock chamber and a high-pressure reactor. The analysis chamber, with a base pressure of $<2 \times 10^{-10}$ torr, is connected to the beamline with a VG Scienta R3000 electron energy analyzer and a twin-anode X-ray source. Before experiment, the catalysts were treated in situ by $H_2$ at 300 °C for 1 h. Then, the catalysts were treated with reaction gas ($H_2/CO = 2$) at 240 °C and 5 bar for 0.5 h, followed by transferring to the analysis chamber under high vacuum for SRPES measurement.

## Computational details

All DFT calculations were performed using the Vienna Ab initio Simulation Package (VASP)[45–48] and the Atomic Simulation Environment (ASE)[49]. The projector augment wave (PAW) method with Perdew-Burke-Ernzerhof (PBE) functional for exchange-correlation term was used with a cutoff energy of 400 eV[50–53]. The Brillouin zone was sampled with a Monkhorst-Pack $1 \times 1 \times 1$ $k$-point for all models[54]. Zero damping DFT-D3 method of Grimme was used to correct van der Waals interactions[55,56]. The models of a $6 \times 6 \times 1$ tri-layer $MoS_2$ supercell and a nanoribbon $MoS_2$ with eight repeating units along ribbon direction, saturated with S monomers were built for simulating in-plane and Mo-edge configurations, respectively, with vacuum region about 15 Å between slabs or ribbons. The energy convergence was set to $1 \times 10^{-5}$ eV in structural optimizations. Fix-bond-length method in ASE was applied to search transition states and a tolerance of 0.06 eV/Å was set for force convergence. The free energies of gaseous molecules were calculated as: $E_{total} + ZPE + H - TS + RT \ln \frac{P}{P_0}$, where $E_{total}$ is DFT calculated total energy, ZPE is the zero-point energy, $H$ and $S$ are enthalpy and entropy based on ideal gas approximation, $T$ is the reaction temperature, $R$ is ideal gas constant (8.314 J mol$^{-1}$ K$^{-1}$) and $P$ and $P_0$ are partial pressure of specific gas components and standard atmosphere pressure, respectively. $E_{total} + ZPE + H_{vib} - TS_{vib}$, where $H_{vib}$ and $S_{vib}$ are enthalpy and entropy parts of non-imaginary vibrations based on harmonic approximation, was implemented to calculate the free energies of reaction intermediates and transition states.

## Reporting summary

Further information on research design is available in the Nature Portfolio Reporting Summary linked to this article.

# Data availability

All data supporting this work are available in the main text and Supplementary Information. Source data are provided with this paper.

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

## Acknowledgements

We acknowledge the financial support from the National Key R&D Program of China (Nos. 2022YFA1504800 to J.H. and L.Y., 2022YFA1504500 to D.D.), the National Natural Science Foundation of China (Nos. 21988101 to D.D., 21890753 to D.D., 22272170 to L.Y., 91945301 to Y.W.), the Strategic Priority Research Program of the Chinese Academy of Sciences (No. XDB36030200 to D.D.), the Fundamental Research Funds for the Central Universities (No. 20720220008 to Y.W. and D.D.). We thank the staff at XAFS beamline (BL14W1) and D-line (BL05U and BL06B1) of the Shanghai Synchrotron Radiation Facilities for assistance with the EXAFS, XANES, and ED-XAS coupling DRIFTS measurements. We thank Dr. Wenguang Yu, Rongtan Li and Dr. Qianru Wang from Dalian Institute of Chemical Physics, Chinese Academy of Sciences for assistance in N$_2$ adsorption measurements, XPS and XRD characterizations, respectively.

## Author contributions

D.D., Y.W. conceived and designed the experiments. J.H. performed the materials synthesis, characterization and performance test. Z.W. and L.Y. contributed to the DFT calculations. K.C. and Q.Z. assisted with data analysis and manuscript revision. Y.Z. conducted the HAADF-STEM tests. R.H. assisted the DRIFTS experiments. M.Z. assisted partial performance test. Y.Q., Y.L., J.M. and X.W. assisted the ED-XAS coupling DRIFTS measurements. J.Z. and L.W. assisted the SRPES experiments. W.W., S.Y., Y.P. and J.Y. assisted the SVUV-PIMS experiments. L.J. and R.S. assisted the data analysis of the EXAFS results. J.H., Z.W., L.Y., Y.W. and D.D. wrote the paper.

## Competing interests

The authors declare no competing interests.
