## [Peer Review File · Nature Communications]

Edge-rich molybdenum disulfide tailors carbon-chain growth for selective hydrogenation of carbon monoxide to higher alcoholsEditorial Note: This manuscript has been previously reviewed at another journal that is not operating a transparent peer review scheme. This document only contains reviewer comments and rebuttal letters for versions considered at *Nature Communications*.

REVIEWER COMMENTS

Reviewer #1 (Remarks to the Author):

I applaud the authors for making additional effort to address all four of my comments. I am happy with the answers and would like to recommend the acceptance of this manuscript.

Reviewer #2 (Remarks to the Author):

The present paper investigated non-noble-metal-based catalysts to achieve high CO conversion and C2-4 alcohol products under relatively milder reaction conditions. The authors tried to explain the effect of edge-enriched MoS₂ catalysts in promoting C₂+ alcohol selectivity and the effect of sulfur vacancies (SVs) in enabling carbon chain growth. Additionally, the effect of a potassium promoter was investigated using various in-situ analyses and DFT calculations. Despite the fact that the authors conducted extensive characterization and theoretical calculations to prove their hypothesis, some additional experiments and analysis are suggested to remove any ambiguity in defining the active sites for the reaction.

1. The authors concluded that the potassium promoter would facilitate the desorption of alcohols through electrostatic interactions with hydroxyls and promote C₂-4OH. Could the authors provide an explanation in the text as to why it selectively desorbed C₂-4 alcohols and not C₁ alcohol? Explain the reason for controlled carbon chain growth on the ER-MoS₂-K catalyst in the text?
2. During the synthesis, the ER-MoS₂ and PD-MoS₂ were treated with different acidic and basic solutions. Does this treatment have any impact on the overall composition of the catalyst? It is recommended that the authors perform XRF or ICP to calculate the metal content after treatment.
3. On page no. 8, authors discussed the formation of SVs with the decrease of Mo oxidation state during XAS analysis and from the integrated intensities of EPR peaks, how the authors are so sure that these are SVs and no MoVs, and that these are formed by the Mo-O-S interactions? How authors identified the location of SVs? Considering that SVs are the primary active sites, it is recommended that the authors confirm the presence of these sites and their location using more convincing techniques.

4. By DFT calculation authors explained the effect of potassium in decreasing the competing activation energies of desorption that hinders the C-O cleavage of the CH_3OH^* or $\text{C}_2\text{H}_5\text{OH}^*$, can the authors confirm this again with the experiments using alcohols as reactants in their reactor system?
5. Similar to supplementary Figure 17, authors are recommended to include activity comparison data for PD-MoS₂ catalysts that shows the effect of potassium addition.
6. Previous experimental and theoretical studies (DFT) have demonstrated that MoS₂ has the ability to facilitate the C-O scission of H_xCO intermediates, leading to a high propensity for hydrocarbon formation. In the current study, it is suggested that hydrocarbons are generated through the decomposition of oxygenates, and the presence of potassium assists in the desorption of oxygenates, therefore decreasing the HC selectivity. However, further elucidation with experimental evidence is needed to provide a more comprehensive explanation.
7. What was the ratio of sulfur to molybdenum (S/Mo) in the catalyst, and would altering this ratio or adjusting the potassium content have any effect on the catalyst's performance?

Responses to the reviewers' comments

We are grateful for all reviewers' in-depth comments and suggestions, which have helped us greatly to improve the quality of this manuscript. We have considered the comments and suggestions seriously, and carried out the related experiments following the suggestions of the reviewers. We have made detailed and **point to point** response and revised the manuscript accordingly with all changes being highlighted in yellow in the revised Manuscript and Supporting Information.

Reviewer #1 (Comments for the Author):

I applaud the authors for making additional effort to address all four of my comments. I am happy with the answers and would like to recommend the acceptance of this manuscript.

Response: We are grateful to the reviewer's recommendation for the acceptance of our manuscript.

Reviewer #2 (Comments for the Author):

The present paper investigated non-noble-metal-based catalysts to achieve high CO conversion and C₂₋₄ alcohol products under relatively milder reaction conditions. The authors tried to explain the effect of edge-enriched MoS₂ catalysts in promoting C₂₊ alcohol selectivity and the effect of sulfur vacancies (SVs) in enabling carbon chain growth. Additionally, the effect of a potassium promoter was investigated using various in-situ analyses and DFT calculations. Despite the fact that the authors conducted extensive characterization and theoretical calculations to prove their hypothesis, some additional experiments and analysis are suggested to remove any ambiguity in defining the active sites for the reaction.

Response: We thank you very much for your valuable comments and suggestions for further improving the quality of our manuscript. We have carried out additional experiments and DFT calculations and have provided corresponding explanations according to your suggestions. The point-by-point responses to the detailed comments and the corresponding revisions are described as follows.

Comment 1. The authors concluded that the potassium promoter would facilitate the desorption of alcohols through electrostatic interactions with hydroxyls and promote C₂₋₄OH. Could the authors provide an explanation in the text as to why it selectively desorbed C₂₋₄ alcohols and not C1 alcohol? Explain the reason for controlled carbon chain growth on the ER-MoS₂-K catalyst in the text?

Response: Thank you for your valuable comments. It seems that there might be a misunderstanding on this point from the reviewer. The decoration of potassium promotes the desorption of both C₂₋₄ alcohols (C₂₋₄OH) and methanol, as described in the main text (page 13, lines 14-20). In the case without the decoration of potassium, the desorption of CH₃OH and C₂₋₄OH requires activation energies of 0.66~0.95 eV (Figs. 4e, f, and Supplementary Fig. 24). In contrast, after decorating the Mo-edge SVs with potassium, the activation energies for the desorption of CH₃OH and C₂₋₄OH are significantly decreased to 0.32~0.45 eV (Figs. 4e, f, and Supplementary Fig. 24). In addition, in-situ DRIFTS characterizations of the CO hydrogenation reaction show gradual enhancement of vibration bands of alcohols (CH₃OH and C₂H₅OH) on the ER-MoS₂-K at elevated temperatures, in contrast to the observation of only alkyl species on the ER-MoS₂ surface under identical conditions, further suggesting the promoted desorption of both CH₃OH and C₂₋₄OH with the decoration of potassium (Supplementary Fig. 26). This is consistent with the remarkable increase in the selectivity of both CH₃OH and C₂₋₄OH over the ER-MoS₂-K catalyst compared with those over the ER-MoS₂ (Supplementary Fig. 22).

The increased selectivity toward C₂₋₄ alcohols over the ER-MoS₂-K catalyst is a result of the promoted carbon-chain growth by edge SVs and the promoted desorption of alcohols by potassium decoration, which both are indispensable. On the one hand, the abundant edge SVs on the ER-MoS₂-K boost carbon-chain growth by facilitating not only the C-O cleavage of CH_xO* for the generation of CH_x* intermediate (page 7, lines 20-27; page 8, last paragraph; page 11, last paragraph), but also the subsequent C-C coupling between the CH_x* and CO* (page 12, lines 4-13). On the other hand, the K decoration promotes the timely desorption of the formed alcohols via electrostatic interaction with the hydroxyl of alcohols. These features enable the controlled carbon-chain growth for the preferential formation of C₂₋₄ alcohols on the ER-MoS₂-K catalyst (page 13, lines 6-28; page 14, lines 1-9). To further clarify these points, we have added more explanations and modified related descriptions in the revised manuscript (page 11, lines 8-9; page 14, lines 12-15).

Comment 2. During the synthesis, the ER-MoS₂ and PD-MoS₂ were treated with different acidic and basic solutions. Does this treatment have any impact on the overall composition of the catalyst? It is recommended that the authors perform XRF or ICP to calculate the metal content after treatment.

Response: Thank you for your comments. According to your suggestion, we have analyzed the metal content of the reduced ER-MoS₂-K and PD-MoS₂-K catalysts by using XRF and ICP (Table R1). Results show that the two catalysts possess similar composition, indicating the negligible impact of the treatment process. We have added the new data in the revised manuscript (page 5, line 3; page 16, lines 25-28; Supplementary Table 2 and Note).

Table R1. Summary of the element composition of the reduced ER-MoS₂-K and PD-MoS₂-K catalysts from XRF and ICP.

Catalyst	Element composition (XRF, wt %) ^a			Element composition (ICP, wt %)	
	Mo	S	K	Mo	K
ER-MoS ₂ -K	56.4	38.6	5.0	43.3	3.8
PD-MoS ₂ -K	56.6	38.4	5.0	41.6	4.3

^a O element was not included in the calculation because the catalysts had been exposed to air before characterization.

Comment 3. On page no. 8, authors discussed the formation of SVs with the decrease of Mo oxidation state during XAS analysis and from the integrated intensities of EPR peaks, how the authors are so sure that these are SVs and no MoVs, and that these are formed by the Mo-O-S interactions? How authors identified the location of SVs?

Considering that SVs are the primary active sites, it is recommended that the authors confirm the presence of these sites and their location using more convincing techniques.

Response: Thank you for your comments and suggestions. For the identification of SVs, in-situ ED-XAS was used to monitor the dynamic change of the Mo chemical state on both ER-MoS₂-K and PD-MoS₂-K catalysts during the H₂ pretreatment. We observed that the absorption edge shifts to higher energies together with an increase in the intensity of its shoulder peak at around 20010 eV (Fig. 3a, b; Supplementary Fig. 12b, c), indicating the decrease of the Mo oxidation state with the exposure of coordinatively unsaturated Mo atoms (*ACS Catal.* 2019, 9, 2568-2579; *Nat. Catal.* 2021, 4, 242-250), which can be ascribed to the formation of SVs due to the removal of surface O atoms and some S atoms from MoS₂ by the H₂ pretreatment. We have also conducted new in-situ mass spectrometry characterizations of the H₂ reduction pretreatment process, in which the signals of H₂O, H₂S, and SO₂ are directly detected over both catalysts (Fig. R1), which also reflects the formation of SVs. EPR spectra, which has been widely used in characterizing SVs on MoS₂ (*Nat. Chem.* 2017, 9, 810-816; *J. Am. Chem. Soc.* 2015, 137, 2622-2627), show the rise of a signal at $g = 2.00$ attributed to SVs, further confirming the existence of SVs on both the reduced ER-MoS₂-K and PD-MoS₂-K catalysts (Supplementary Fig. 14a). In addition, if Mo atoms were removed and Mo vacancies were formed, it will increase the S/Mo ratio in the local area around the Mo vacancies, which will lead to increased oxidation state of Mo rather than decreased oxidation state of Mo. Thus, our results suggest the formation of sulfur vacancies during the H₂ reduction pretreatment.

The removal of surface oxygen is also evidenced by in-situ DRIFTS combined with in-situ ED-XAS characterizations of the process, which display the decreased intensity of the absorption peaks for the vibrations of Mo=O and Mo-O-Mo at around 930 cm⁻¹ and 760 cm⁻¹, respectively (*J. Phys. Chem.* 1996, 100, 14144-14150) (Fig. 3c, d, and Supplementary Fig. 12d, e). Our previous study has demonstrated that oxygen adsorbed at the edge and in-plane S vacancies are primarily in Mo=O and Mo-O-Mo configurations, respectively (*Nat. Catal.* 2021, 4, 242-250) (Fig. R2a). Over the ER-MoS₂-K catalyst, only the removal of Mo=O species during H₂ pretreatment was obviously observed (Fig. 3c, d), suggesting that the ER-MoS₂-K catalyst is enriched with edge SVs. In contrast, the PD-MoS₂-K catalyst possesses more in-plane SVs as reflected by the observation of the removal of Mo-O-Mo species during H₂ pretreatment (Supplementary Fig. 12d, e). We have also conducted new in-situ O₂-adsorption DRIFTS characterizations of the K-free MoS₂ samples, which show the rise of mainly Mo=O species on ER-MoS₂ while mainly Mo-O-Mo species on PD-MoS₂, also suggesting the totally different SVs distributions on the two kinds of catalysts (Fig. R2b, c). We have added the new data and the references with related discussion in the revised manuscript (page 8, lines 6-12, lines 19-25; Supplementary Figs. 13 and 16).

Fig. R1. In-situ MS detection of the H₂ pretreatment products on ER-MoS₂-K (a) and PD-MoS₂-K (b).

Fig. R2. (a) DFT-calculated structures of the adsorption of oxygen at the edge (top) and in-plane (bottom) S vacancies in the reference (*Nat. Catal.* 2021, 4, 242-250). (b and c) In-situ DRIFTS spectra of the ER-MoS₂ (b) and PD-MoS₂ (c) during the O₂ adsorption process.

Comment 4. By DFT calculation authors explained the effect of potassium in decreasing the competing activation energies of desorption that hinders the C-O cleavage of the CH₃OH* or C₂H₅OH*, can the authors confirm this again with the experiments using alcohols as reactants in their reactor system?

Response: Thank you for your kind suggestion. Accordingly, we have performed the hydrogenation of both CH₃OH and C₂H₅OH over the ER-MoS₂ and ER-MoS₂-K catalysts. Results show that the ER-MoS₂ catalyst

presents significantly higher production rates for the CH₄ and C₂H₆ from CH₃OH and C₂H₅OH hydrogenation, respectively, compared with those over the ER-MoS₂-K catalyst (Fig. R3a). However, the apparent activation energies for the hydrogenation of CH₃OH to CH₄ and C₂H₅OH to C₂H₆ over the ER-MoS₂ catalyst are similar to those over the ER-MoS₂-K catalyst (Fig. R3b, c), being consistent with our DFT calculations that potassium modification has little effect on the C-O cleavage activity (Fig. 4e and Supplementary Fig. 24). Combined with the in-situ high pressure CO hydrogenation DRIFTS characterizations, showing the gradually enhanced vibration bands of alcohols on the ER-MoS₂-K at elevated temperatures in contrast to that showing only alkyl species on the ER-MoS₂ (Supplementary Fig. 26), the effect of potassium in promoting the desorption of alcohols can be further validated. We have added the new data with related discussion in the revised manuscript (page 13, lines 24-27; page 16, lines 3-6; Supplementary Fig. 25 and Note).

Fig. R3. (a) Formation rate of CH₄ and C₂H₆ via the hydrogenation of CH₃OH and C₂H₅OH, respectively, over the ER-MoS₂ and ER-MoS₂-K catalysts at 240 °C. (b, c) Arrhenius plots calculated based on the reaction rate for the hydrogenation of CH₃OH (b) and C₂H₅OH (c) over ER-MoS₂ and ER-MoS₂-K. Catalysts were pretreated in-situ by H₂ at 300 °C for 1 hour before reaction.

Comment 5. Similar to supplementary Figure 17, authors are recommended to include activity comparison data for PD-MoS₂ catalysts that shows the effect of potassium addition.

Response: Thank you for your suggestion. We have conducted additional performance test for the PD-MoS₂ catalyst. Results further confirm the effect of potassium addition in enhancing the selectivity toward alcohols (Fig. R4). We have added the new data in the revised manuscript (page 13, line 7; Supplementary Fig. 23).

Fig. R4. Distribution of CO hydrogenation products over PD-MoS₂ (a) and PD-MoS₂-K (b) at different reaction temperatures. The product selectivity was calculated on a CO₂-free basis. Catalysts were pretreated in-situ by H₂ at 300 °C for 1 hour before reaction. Reaction activity tests were performed at 50 bar, 3000 mL g_{cat.}⁻¹ h⁻¹ and H₂/CO ratio of 2.

Comment 6. Previous experimental and theoretical studies (DFT) have demonstrated that MoS₂ has the ability to facilitate the C-O scission of H_xCO intermediates, leading to a high propensity for hydrocarbon formation. In the current study, it is suggested that hydrocarbons are generated through the decomposition of oxygenates, and the presence of potassium assists in the desorption of oxygenates, therefore decreasing the HC selectivity. However, further elucidation with experimental evidence is needed to provide a more comprehensive explanation.

Response: Thank you for your comment. As mentioned above, we have performed the hydrogenation of both CH₃OH and C₂H₅OH over the ER-MoS₂ and ER-MoS₂-K catalysts, which all proceed with considerable reaction rates for the formation of CH₄ and C₂H₆ (Fig. R3a), suggesting that hydrocarbons can be generated through the hydrogenation of alcohols. With the decoration of potassium, formation rates of CH₄ and C₂H₆ on the ER-MoS₂-K catalyst are significantly suppressed (Fig. R3a). As discussed in the response to Comment 4, the ER-MoS₂ and ER-MoS₂-K catalysts present similar activities for C-O cleavage (Fig. R3b, c, Fig. 4e, Supplementary Fig. 24), but different abilities in desorbing alcohols (Fig. 4e, Supplementary Figs. 24 and 26). These results reveal that potassium modification can improve the alcohols selectivity by promoting the desorption of alcohols. We have added the new data with related discussion in the revised manuscript (page 13, lines 24-27; page 16, lines 3-6; Supplementary Fig. 25 and Note).

In addition, we have also calculated the activation energy for the C-O dissociation of CH₃O* on Mo-edge SVs, which is 1.45 eV and surmountable at 240 °C but is higher than the activation energy for the hydrogenation

of CH_3O^* to CH_3OH^* (1.05 eV). This suggests that the C-O cleavage of CH_3O^* intermediates may partially contribute to the formation of hydrocarbons but with limited amount due to the relatively higher energy barrier.

Comment 7. What was the ratio of sulfur to molybdenum (S/Mo) in the catalyst, and would altering this ratio or adjusting the potassium content have any effect on the catalyst's performance?

Response: Thank you for the comments. According to the XRF results as shown in Table R1, the ratios of S/Mo in the reduced ER-MoS₂-K and PD-MoS₂-K catalysts are calculated as 2.05 and 2.04, respectively, which are close to the S/Mo ratio of standard MoS₂. During the H₂ reduction of the catalyst, a small amount of S atoms was removed for the formation of SVs (page 8, lines 1-25). This could alter the S/Mo ratio but only within a limited extent, since the two-dimensional MoS₂ structure is still well-maintained. Newly conducted performance tests show that the H₂ reduction process markedly improves the CO conversion and C₂₋₄OH selectivity over the ER-MoS₂-K catalyst (Fig. R5), suggesting that the formation of SVs with slight alteration in the S/Mo ratio is significant for improving the catalytic performance of MoS₂-based catalysts (page 8, lines 14-15; Supplementary Fig. 15).

Fig. R5. Catalytic performances of the ER-MoS₂-K catalyst before (fresh) and after (reduced) H₂ pretreatment at 300 °C for 1 h.

To illustrate the effect of potassium content, we have conducted additional performance tests over the ER-MoS₂-K catalysts with different potassium content (Fig. R6). Results show that the alcohols selectivity increases with increasing potassium content, and the highest selectivity and formation rate of C₂₋₄OH are achieved at a K/Mo mole ratio of 0.2. We have added the new data with related discussion in the revised manuscript (page 5, line 5; Supplementary Fig. 4 and Note).

Fig. R6. **(a)** Distribution of CO hydrogenation products over the ER-MoS₂-K catalysts with different potassium content (x is the mol ratio of K/Mo). The product selectivity was calculated on a CO₂-free basis. **(b)** Evolutions of net formation rate of methanol and C₂₋₄ alcohols with increasing K/Mo ratio. Catalysts were pretreated in-situ by H₂ at 300 °C for 1 hour before reaction. Reaction activity tests were performed at 50 bar, 240 °C, 3000 mL g_{cat.}⁻¹ h⁻¹ and H₂/CO ratio of 2.

REVIEWERS' COMMENTS

Reviewer #2 (Remarks to the Author):

The paper seems to be properly revised to reply to referee's comments. I believe the paper is acceptable as it is.

Responses to the reviewers' comments

Reviewer #2 (Comments for the Author):

The paper seems to be properly revised to reply to referee's comments. I believe the paper is acceptable as it is.

Response: We are grateful to the reviewer's recommendation for the acceptance of our manuscript.